# Gap Junctions and Hemichannels Composed of Connexins and Pannexins Mediate the Secondary Brain Injury Following Intracerebral Hemorrhage

**DOI:** 10.3390/biology11010027

**Published:** 2021-12-25

**Authors:** Yan Zhang, Suliman Khan, Yang Liu, Rabeea Siddique, Ruiyi Zhang, Voon Wee Yong, Mengzhou Xue

**Affiliations:** 1Department of Cerebrovascular Diseases, The Second Affiliated Hospital of Zhengzhou University, Zhengzhou 450014, China; pudingww@163.com (Y.Z.); suliman.khan18@mails.ucas.ac.cn (S.K.); liuy@zzu.edu.cn (Y.L.); rabeeakhan11@gmail.com (R.S.); zhangrui1994@gmail.com (R.Z.); 2Academy of Medical Science, Zhengzhou University, Zhengzhou 450001, China; 3Hotchkiss Brain Institute and Department of Clinical Neurosciences, University of Calgary, Calgary, AB T3A 4X9, Canada

**Keywords:** intracerebral hemorrhage, secondary brain injury, pannexin, hemichannel, inflammation, oxidative stress

## Abstract

**Simple Summary:**

Intracerebral hemorrhage (ICH) is a leading medical problem without effective treatment options. The poor prognosis is attributed to the primary brain injury of the mechanical compression caused by hematoma, and secondary brain injury (SBI) that includes inflammation, glutamate excitotoxicity, oxidative stress and disruption of the blood brain barrier (BBB). Evidences suggests that gap junctions and hemichannels composed of connexins and pannexins regulate the inflammation and excitotoxicity insult in the pathological process of central nervous system disease, such as cerebral ischemia and neurodegeneration disease. In this manuscript, we discuss the fact that connexins- and pannexins-based channels could be involved in secondary brain injury of ICH, particularly through mediating inflammation, oxidative stress, BBB disruption and cell death. The details provided in this manuscript may help develop potential targets for therapeutic intervention of ICH.

**Abstract:**

Intracerebral hemorrhage (ICH) is a devastating disease with high mortality and morbidity; the mortality rate ranges from 40% at 1 month to 54% at 1 year; only 12–39% achieve good outcomes and functional independence. ICH affects nearly 2 million patients worldwide annually. In ICH development, the blood leakage from ruptured vessels generates sequelae of secondary brain injury (SBI). This mechanism involves activated astrocytes and microglia, generation of reactive oxygen species (ROS), the release of reactive nitrogen species (RNS), and disrupted blood brain barrier (BBB). In addition, inflammatory cytokines and chemokines, heme compounds, and products of hematoma are accumulated in the extracellular spaces, thereby resulting in the death of brain cells. Recent evidence indicates that connexins regulate microglial activation and their phenotypic transformation. Moreover, communications between neurons and glia via gap junctions have crucial roles in neuroinflammation and cell death. A growing body of evidence suggests that, in addition to gap junctions, hemichannels (composed of connexins and pannexins) play a key role in ICH pathogenesis. However, the precise connection between connexin and pannexin channels and ICH remains to be resolved. This review discusses the pathological roles of gap junctions and hemichannels in SBI following ICH, with the intent of discovering effective therapeutic options of strategies to treat ICH.

## 1. Introduction

ICH refers to blood entering into the brain parenchyma, ventricle system, or subarachnoid space from a rupturing cerebral vessel. ICH causes a huge burden to patients and to society. The mortality rate in ICH patients is approximately 50%, while most of the survivors lose the capability of living independently [1,2]. Notably, secondary brain injury (SBI), which refers to inflammation [3], oxidative stress [4], cerebral vasospasm [5], and blood brain barrier (BBB) hyperpermeability [6], further drives brain cell death. SBI is a major cause of the high mortality and morbidity in ICH patients. Until now, only limited treatment options are available for ICH, therefore novel therapeutic options are needed to be developed [7,8,9].

Gap junctions and hemichannels composed of connexins (Cx) play a crucial role in the ICH pathology. A hemichannel or connexon comprised of six connexin proteins allows approximately 1kDa sized molecules to pass through the membrane. These molecules include metabolites, ATP, ADP, cAMP, Ca^2+^, nutrients, second messengers, and adenosine. The connexons on the membrane are docked with a connexon on a neighboring cell resulting in a gap junction channel, which then allows passage of ions into adjacent cells [10] in order to modulate the communications between cells or the extracellular environment and intracellular environment [11]. The gating of hemichannels and gap junctions is regulated by different factors, such as trans-junctional/transmembrane voltage, changes of extracellular and intracellular Ca^2+^, mechanical stimulation, changes in phosphorylation, reactive oxygen species (ROS), and nitrosylation [12]. Moreover, pannexins are also important channels that can form structurally similar channels to connexins. Pannexins are permeable to molecules below 900 Da, such as nucleotides and ions [13]. At least one of the three known proteins, Panx1, Panx2, and Panx3, has been found in every organ in mammals.

Previous evidence has shown that pannexin and connexin play a critical role in the pathological process of cerebrovascular diseases, such as cerebral ischemia [14] and neurodegeneration disease [15], mainly through regulating inflammatory and excitotoxic insults. The precise role of connexin and pannexin channels in ICH has not been well understood yet. The aim of our current review is to outline the possible roles of connexin and pannexin channels in SBI following ICH.

## 2. The Expression and Functions of Connexins and Pannexins in the Brain

Connexins and pannexins are abundantly expressed in the CNS. All neurons express Cx36 and Cx45; astrocytes, the most numerous cells in CNS are coupled via gap junctions, which primarily express Cx43 and Cx30; microglia predominantly express Cx46, Cx32, Cx36, and Cx43. Oligodendrocytes (the myelinating cells of the CNS) are known for their ability of extensive connectivity with astrocytes, as well as with each other, and they chiefly express Cx29, Cx32, and Cx47 [16]. Endothelial cells mainly express Cx37 and Cx40 [17]. Until now, three pannexins have been identified, namely Panx1, Panx2, and Panx3, with Panx1 and Panx2 being widely expressed in the CNS [15].

### 2.1. The Expression and Function of Gap Junction Channels in the Brain

Under physiology conditions, gap junction channels between brain cells exert critical functions in maintaining the brain’s homeostasis. Gap junctions between neurons form electrical synapses, which play an important functional role in brain cell synchronization [18]. On the other hand, carbenoxolone, a general gap junction blocker, inhibits spikelets induced by local stimulation of stratum oriens, thereby decreasing the rate of firing antidromic action potential, which contributes to early events generation in the hippocampus of neonates [19]. Besides, Cx36 (neuronal gap junction protein) increases the junctional conductance by 10-fold or even more within ten minutes after cell break-in with pipettes [20]. Moreover, neuronal gap junctions induce memory and play an important role in learning.

Oligodendrocytes and astrocytes form few gap junctions essential for homeostatic and nutrient support. Specific transporters allow glucose molecules to enter from blood to the astrocytic end-feet. These glucose molecules are then transported from one astrocyte to another through Cx43/Cx30 gap junctions. After their release in the proximity of neurons, uptake occurs. In addition, gap junctions coupled with the astrocyte network are essential for extracellular potassium ion (K^+^) clearance. After uncoupling of the astrocytes, hampering of K^+^ clearance takes place, while extracellular K^+^ accumulation causes hyperexcitability of neurons. Accumulating evidence indicates that the defective control of ion/fluid exchanges of astrocytes may cause brain edema, fluid cysts, and astrocyte/myelin vacuolation [21]. Furthermore, Cx30 and Cx43 of the astrocytes are normally distributed in perivascular end-feet; Cx30 and Cx43 double knockout of astrocytes induce end-feet swelling [22].

While confronting pathological insults, connexin-based gap junction channels may exhibit opposite roles depending on the disease conditions. A substantial body of evidence indicates that gap junction channels may mediate cell death through a mechanism termed ‘bystander death’ or the ‘kiss of death’, which describes the death signal that propagates to neighboring cells through gap junctions [12]. Evidence is accumulating that Ca^2+^ ions are the most probable cell-killing signals spread through gap junctions [23]. Moreover, except for the large molecules of cytochrome C, apoptosis inducer Ca^2+^ and IP3 have also been shown to spread via gap junctions to exacerbate cell apoptosis cascades. Earlier research had found that upregulation of the proto-oncogene b-cell leukemia/lymphoma 2 (Bcl2) increases resistance to injury, however, the gap junction formation with vulnerable cells leads to compromised resistance to metabolic inhibition, calcium overload, and oxidative stress [24]. Knockdown Cx36-containing gap junctions prevent neuronal death from ischemia [25], while over-expression of Cx36 results in increased neuronal death [25]. Consistent with this phenomenon, the inhibition of the gap junction (astrocytic) increases the protective effects following cerebral ischemia [26], while the gap junction blockers modulate seizure-linked behavioral parameters [27].

By contrast, several investigations show that gap junctions can attenuate brain cell death by distributing intracellular toxic substances to healthy neighbors. Besides, potential ‘rescue messengers’ such as ATP, energy molecules, and reduced glutathione, and ascorbic acid can also flow through gap junctions in order to induce cell survival [12]. Gap junctional intercellular communication (GJIC) promotes reversal of ischemia-mediated hippocampal apoptosis and cognitive impairment. While GJIC suppression facilitates hippocampal apoptosis and cognitive impairment [28].

These results suggest that the occurrence of two-way traffic between injured cells possibly depends on the different stages of pathological process and signal-receiving cellular environment.

### 2.2. The Expression and Function of Connexin and Pannexin Hemichannels in the Brain

Connexin and pannexin mediated communications of neurons widely occur during brain development. In late prenatal neurogenesis, gap junctions can control the migration and division of precursor cells [29]. Connexin hemichannels are thought to remain closed, primarily while pannexin channels are active under normal physiological conditions. However, the pannexin channels would contribute to ATP release and the connexin undocked hemichannel may take on a prolonged or more frequently open state following a pathology insult. These functions may further induce cell death through depolarization of the membrane, loss of small metabolites, cytoplasmic Ca^2+^ elevation, and the ionic gradients’ collapse [30].

It has been proposed that inflammation is associated with the opening of unopposed hemichannels [31]. ATP released from hemichannels leads to autocrine activation of purinergic P2X receptors (P2X). P2X7 then activates pyrin domain-containing-3 (NLRP3) inflammasome, the nucleotide-binding oligomerization domain-like receptor, and enhances the inflammatory effect, which then leads to increased Cx43 expression, opening hemichannels, as well as higher ATP release [32]. Thus, blockade of hemichannel expression or inhibiting P2X7 receptors during neuroinflammation might prevent neuronal damage [33]. Hemichannels also release glutamate, which is known as a paracrine messenger of cell death [34]. Orellana et al. pointed out that glutamate and ATP released by astroglial Cx43 hemichannels activate Panx1 hemichannels, which facilitate neuronal death [35]. Moreover, cell death also was accelerated by the metabolic inhibition of astrocytes through the opening of Cx43 hemichannel [36]. Hence, the abnormal opening of connexin and pannexin hemichannels would predominantly lead to detrimental effects in the brain.

## 3. Connexin and Pannexin Channels in ICH

The poor prognosis of ICH may attribute to the primary brain injury that, due to the mechanical compression, caused by hematoma and second brain injury, mainly including inflammation and oxidative stress, which is mediated through a series of events induced by primary injury and releasing of clot components [2]. When a blood vessel in the brain ruptures, local cerebral blood flow and cerebral perfusion pressure may drop while intracranial pressure increases, which would induce cerebral vasospasm and cerebral ischemia [37]. While blood leaks into the brain parenchyma, the hematoma and the degradation products of erythrocytes (such as hemoglobin, heme and iron), and complement components (mainly C3a and C5a) activate microglia. The invasion of neutrophils may lead to the release of toxic substances such as thrombin, ROS and matrix metalloproteinases (MMPs). Collectively, the result is neuroinflammation, neuronal and glial cell death, vasogenic edema, and further breakdown of the BBB [38]. Based on the impact of connexin and pannexin channels in brain pathology discussed above, the subsequent paragraphs will explore the effects of connexin and pannexin channels in SBI following ICH.

### 3.1. Connexin and Pannexin Channels Are Implicated in Neuroinflammation Following ICH

Following ICH or primary injury, a proinflammatory cascade in the peri-hematoma induces neural cell death. This proinflammatory cascade is composed of infiltrated leucocytes, activated astrocytes, and microglia [3,4,5,6,7,8,9,10,11,12,13,14,15,16,17,18,19,20,21,22,23,24,25,26,27,28,29,30,31,32,33,34,35,36,37,38,39], and is associated with the activity of connexin channels.

During the development and progression of ICH, hematoma degradation products (hemin and hemoglobin) activate astrocytes [40], which are then accumulated in the surrounding area and release chemokines and cytokines to induce inflammatory responses, neuronal apoptosis and destruction of BBB [41]. Further inflammatory factors are released by the excessive opening of Cx43 hemichannels, which are present on the surface of reactive astrocytes, in order to aggravate the inflammatory response and activate the immune system at the damaged area [42]. In the ICH model inflicted by collagenase IV injection into the brain, Cx43 upregulation and excessive Cx43 hemichannel opening have been observed [43]. Furthermore, downregulated Cx43 and nuclear translocated YAP play a role in hemoglobin-activated astrocytes and are interdependent, such that reducing Cx43 induces YAP nuclear translocation [44].

Microglia are key players in ICH mediated inflammatory responses [45]. Microglia can be activated by hematoma degradation products and fibrinogen in the hematoma [46]. Activated microglia release IL−1β, IL−6 and TNF-α molecules in order to propagate inflammatory responses and promote the recruitment of leukocytes. They also facilitate the release of chemokines such as CCL2, CCL5, CXCL8 in order to attract monocytes, lymphocytes, and neutrophils [38]. The expression levels of Cx43 and Cx36 are comparatively very minimally produced during the resting surveillance state of microglia. However, these expression levels are increased after the activation of microglia by pro-inflammatory conditions. More importantly, the released proinflammatory cytokines from activated microglia impede GJIC and activate Cx43 hemichannels in astrocytes [47] (Figure 1). Extracellular ATP released by the opening of hemichannels leads to the activation of purinergic receptors P2X7R and P2X4R, which can further activate NRLP3 pro-inflammatory inflammasome pathway in microglia to exacerbate inflammation responses [48] (Figure 1). Hence, connexin based channels play an important role in inflammatory injury mediated by activated microglia, and purinergic signaling would be partly involved in determining the activation state of microglia. Besides, Cx43 hemichannels mediated release of gliotransmitters (ATP/glutamate) may also result in the opening of Panx1 channels and Cx36 hemichannels in neurons. Consequent neuronal Ca^2+^ overload can lead to numerous deleterious consequences, including structural neuronal alterations and increased oxidative stress [30] (Figure 1).

At the same time, gap junction channels can be regulated by pro-inflammatory cytokines. Pro-inflammatory cytokines and ATP (10–100 µM) induce a rapid and concentration-dependent inhibition of GJIC in cultured cortical astrocytes [49]. IL−1β in combination with TNF leads to a robust decrease in astroglial coupling [50]. Moreover, IL−1β treatment also reduces wave propagation of calcium between astrocytes [51]. Pro-inflammatory cytokines mediated GJIC inhibition reduces death signal molecules or the spread of toxic substances, thereby preventing impairment and survival of neurons. Correspondent with this idea, astroglial GJIC contributes to the propagation of death signals partly by activating p38/stress-activated protein kinase 2 following CNS injury [52].

### 3.2. Connexin Channels Mediate the Permeability of BBB Following ICH

The integrity of BBB is maintained by gap junctions, tight junctions, and adherens junctions, which constitute a unique junction complex. The expression of connexin-based channels takes place throughout the BBB as these connexins are produced by pericytes and endothelial cells [6]. Evidence suggests that connexin-mediated GJIC regulates the integrity and maintains the normal function of the BBB [14]. Non-specific pannexin and connexin blockers inhibit the barrier’s function, suggesting that these molecules induce the progression of ICH, which disrupts the BBB, as well as increase permeability in paracellular and transcellular routes, thereby leading to vascular leakage. Recently, Cx43 upregulation and excessive Cx43 hemichannel opening were observed in mice after ICH injury [43], which may confer detrimental barrier permeability. Moreover, connexin hemichannels may increase BBB permeability by releasing ATP and glutamate after ICH. Allison et al. demonstrated that Cx43 gap junctions play a crucial role in the hyperpermeability of the endothelial barrier by modulating the structure of tight junction [53]. Similar to this finding, another group showed that Cx36 may interact with the PDZ domain-containing protein zonula occludens-1 (ZO-1), ZO-2 and ZO−3 [54]. Further investigations confirmed that increasing *p*-Cx43 altered the integrity of the BBB through the activation of the PI3K and ERK pathways in recombinant tissue plasminogen activator (rtPA)-associated brain ischemia hemorrhagic transformation [55]. Since propagation of Ca^2+^ affects the integrity of cytoskeleton or endothelial cell function, therefore elevated GJIC is thought to be pathogenic to the function of BBB [56]. Besides, the excessive opening of Cx43 hemichannels following hypoxia would directly mediate BBB hyperpermeability by resulting endothelial cell death [57].

### 3.3. Connexin Channels Are Involved in Oxidative Stress Following ICH

Following ICH progression, the accumulation of ROS induces oxidative stress, which contributes to SBI caused by inflammation and BBB disruption [4]. Interestingly, ROS/RNS are involved in regulating the change of connexin and pannexin-based channel properties, both in systemic vasculature and brain cells [58,59]. Moreover, oxidation products were shown to increase hemichannel activity whilst reducing GJIC [60]. Thus, antioxidant therapy may inhibit hemichannel activity [61] and attenuate apoptosis [62]. As mentioned above, Cx43 gap junctions between astrocytes essentially retain homeostasis. The absence of expression or its channels blockage induces ROS-mediated death of astrocytes, and Cx43-mediated gap junctions in astrocytes positively affect oxidative stress resistance [60]. Recent studies have found that astrocyte mediated apoptosis disrupts homeostasis simultaneously, as well as downregulated Cx43, which in turn lead to the conformational changes in Cx43, closing and degradation of the channels. These can further prevent Nrf2 nuclear translocation and protein kinase C alpha (PKCα) phosphorylation to antioxidant stress. Cx43 helps liberate Nrf2 from Kelch-like ECH-associated protein 1, as well as allows the nuclear translocation in order to promote genes encoded for phase II detoxification enzymes, which are involved in processes of anti-apoptosis and antioxidant stress [63] (Figure 2). According to these findings, it seems that hemichannels and gap junctions in astrocytes may exert a beneficial effect on oxidative stress prevention through Nrf2 regulation.

### 3.4. Connexin Channels Take Part in Regulating Cerebral Vasospasm Following ICH

Cerebral vasospasm, a severe complication of subarachnoid hemorrhage [64], is a well known condition, in which vasomotion requires gap junctions for intercellular communications. Cx 37 and 40 are mainly expressed by endothelial cells in healthy conduit arteries, while Cx43 and Cx45 are expressed by vascular smooth muscle cells [17]. Connexin channels are thought to be involved in the process of cerebral vasospasm, which is associated with subarachnoid hemorrhage. Increased phosphorylation of Cx43 via the p38MAPK and protein kinase C (PKC) pathways was found to mediate the development of cerebral vasospasm in animal models [65,66]. Moreover, experimental cerebral vasospasm is attenuated by gap junction blockers, which also down-regulate overexpressed Cx43 protein in subarachnoid hemorrhage [67]. However, Lan et al. showed a marked decrease in Cx40 after subarachnoid hemorrhage, upregulation of Cx40 mediated by nitric oxide attenuated cerebral vasospasm via the nitric oxide-cyclic guanosine monophosphate-protein kinase G pathway after subarachnoid hemorrhage [17]. Thus, different connexin subtypes may play various roles in vasospasm following subarachnoid hemorrhage.

### 3.5. Hemichannels Are Involved in Cell Death Following ICH

The local concentration of glutamate is highly elevated following ICH, which subsequently overstimulates N-methyl-D-aspartate receptors (NMDARs) to increase intracellular Ca^2+^ and neuronal death [68]. Current evidence indicates that Panx1 channels can be activated by NMDARs via Src family kinases to induce neuronal apoptosis [69]. Furthermore, in rat primary spinal neurocytes, Panx1 overexpression enhances signaling of intracellular Ca^2+^, as well as upregulates Bax (apoptotic protein) levels, and apoptosis pathway proteins, including cleaved caspase-3 and PARP-1; conversely, Panx1 depletion reversed the pro-apoptosis effect [70]. It is important to highlight that the Panx1 is linked with arresting the metabolic activity of apoptotic cells through facilitating intracellular ATP loss [71]. Accordingly, Zhou and colleagues recently demonstrated that the expression of Panx1 significantly increased after ICH, while the peak level was noted at 48h post-ICH [72].

## 4. Strategies Target Gap Junctions and Hemichannels Following ICH

A growing body of evidence showed that hemichannels blocking by pharmaceutic preparation or genetic measures may improve neurological function following ICH. Gap19, a Cx43 mimetic peptide, significantly alleviated hematoma volume and neurological deficits after ICH injury by downregulating Cx43 and regulating YAP inflammation signaling in astrocytes [43]. Consistent with this result, multiple Cx43 inhibitors (eg, carbenoxolone and dynasore) were demonstrated to decrease hematoma volume and BBB disruption in ICH mice [73]. Most importantly, carbenoxolone, also as a Panx1 inhibitor, remarkably improved cognitive function, reduced brain edema and BBB injury in rats post-ICH. Moreover, it reduced the degenerative Fluoro-Jade B positive cells, inhibited caspase3 activation and attenuated TUNEL positive cells in the proximity of ICH hematoma [72]. In another investigation, carbenoxolone was reported to down-regulate Cx43 protein expression and attenuate the experimental cerebral vasospasm after subarachnoid hemorrhage in rabbits [67].

For a long time, GJIC was deemed to amplify the extent of injury by transmitting the signals of apoptotic and necrotic cells. Thus, blockade of gap junctions has been shown to provide neuroprotection in CNS diseases, such as cerebral ischemia [74]. However, administration of gap junction inhibitors, octanol and carbenoxolone, failed to attenuate the neurological deficits induced by subarachnoid hemorrhage, and they did not reduce neuronal apoptosis. On the contrary, carbenoxolone increased post subarachnoid hemorrhage mortality and exacerbated its apoptosis [75]. Similar results have been found in ICH mice, where a high dose of carbenoxolone aggravated neurological deficits and increased mortality 72 h after the treatment [76]. Moreover, carbenoxolone treatment deteriorated barrier permeability after ICH [76].

These pieces of evidence suggest that gap junction and hemichannel inhibitors may play a diverse role in different models and periods of ICH. In addition, the dosage of the medication is also an important factor affecting its neuroprotective effect.

## 5. Conclusions

This review summarized evidence for the roles that connexin and pannexin channels appear to play in ICH. Based on the data, we infer that connexin and pannexin-based channels are involved in the process of inflammation, oxidative stress, BBB disruption and cell death in ICH. Notably, the inhibition of both pannexin channel and connexin hemichannels appears beneficial following ICH, whereas the role of gap junction channel function is ambiguous, since gap junction channel uncoupling appears to be detrimental in BBB dysfunction and oxidative stress, but to be beneficial in reducing inflammatory responses. These data suggest that regulating gap junctions and hemichannels may lead to new therapeutic strategies against secondary brain injury induced by ICH. However, the most specific blockers currently available are mimetic peptides with sequences very similar to that of the extracellular loop of connexins. Therefore, specific channel blockers targeting certain connexin or pannexin monomer subtypes are urgently required. It is obvious that studies focusing on the association between connexin channels and ICH are sporadic and far from complete. The precise mechanism of connexin and pannexin channels to different metabolic products of hematoma remains indistinct, and further investigations are required to explore the temporal and spatial characteristics of connexin and pannexin channel responses after ICH. It is worth noting that understanding the precise pathologic roles of connexin and pannexin channels in ICH would offer better therapeutic options to reduce the dismal consequences of ICH.

## Figures and Tables

**Figure 1 biology-11-00027-f001:**
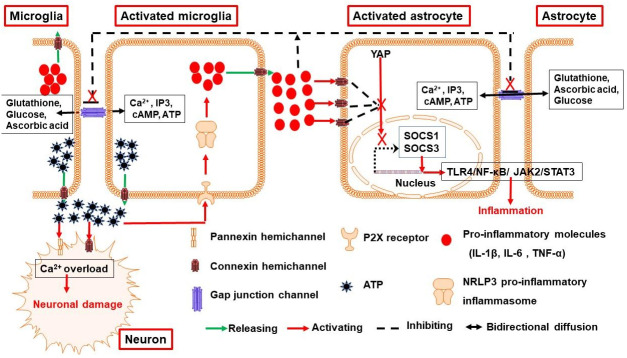
Possible mechanisms by which connexins and pannexins mediates inflammatory responses following ICH. Microglia activated by hematoma degradation products release pro-inflammatory molecules, including IL−1β, IL−6, TNF-α and ATP. These molecules may auto-activate P2X receptors on the membrane of microglia, leading to the activation of NRLP3 pro-inflammatory inflammasome to exacerbate the release of pro-inflammatory molecules. Conversely, they also activate Cx43 hemichannels in astrocytes to aggravate the inflammatory responses by inactivating pro-inflammatory YAP-SOCS1-SOCS3-TLR4-NFκB and JAK2-STAT3 axis pathways. Furthermore, ATP released from Cx43 hemichannels results in the opening of Cx36 hemichannel and Panx1 channels in neurons, eventually causing neuronal damage. ICH, intracerebral hemorrhage; Cx, connexins; Panx, pannexins; P2X, purinergic P2X receptors; NRLP3, nucleotide-binding oligomerization domain-like receptor family, pyrin domain-containing−3; HCs, hemichannels; GJIC, gap junctional intercellular communication.

**Figure 2 biology-11-00027-f002:**
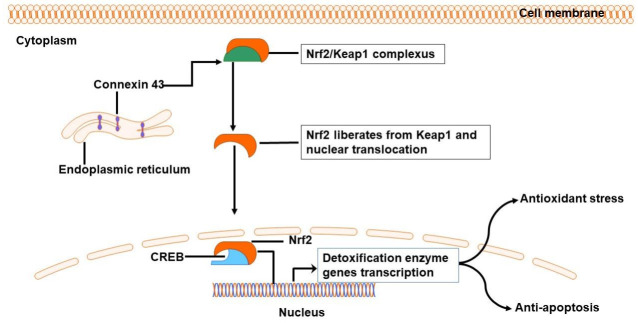
Possible mechanism by which astrocyte Cx43 resists oxidative stress following ICH. Cx43 from endoplasmic reticulum helps liberate Nrf2 from Keap1 and allows its nuclear translocation, promoting phase II detoxification enzyme genes involved in antioxidant stress and anti-apoptosis. ICH, intracerebral hemorrhage; Cx43, connexin 43; OS, oxidative stress; Nrf2, nuclear factor erythroid 2-related factor 2; CREB, cAMP-response element-binding protein.

## Data Availability

Please contact the corresponding author to discuss the availability of the data and materials.

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
