# Peer review of "Gap Junctions and Hemichannels Composed of Connexins and Pannexins Mediate the Secondary Brain Injury Following Intracerebral Hemorrhage"

_biology, 2021, doi:10.3390/biology11010027_

Round 1

Reviewer 1 Report

This is a well written and comprehensive review on the topic of connexins and pannexins in brain injury. The review is to be included in a special issue on gap junctions and connexins and thus the topic is appropriate. The figured provided with the review are a nice addition and are well constructed.  A few comments are noted:

Please add Fibrinogen as an activator of microglia after blood leaks into brain (heading 3). Add references from Akassoglou group, for example “Fibrinogen-induced perivascular microglial clustering is required for the development of axonal damage in neuroinflammation” Nature Communications 2012.

Please add an additional section prior to the conclusions where you discuss potential treatment strategies or therapeutic approaches. The review is comprehensive of studies performed, and it would be great if authors could summarize the field moving forward from a therapeutic perspective. Authors could also give their opinion for future drug strategies. Drug studies completed could also be added to this section, some moved from the body of the manuscript and discussed in this section. For example, numerous drugs targeting Cx43 have been explored, some discussed already and several others worth noting.

The manuscript needs proof-reading for grammatical English language errors.

Reviewer 2 Report

Overall, I think this is an important topic to be reviewing.  And not an easy one, at that.  Overall there are intermittent times when it is hard to read because of grammatical issues and there are contradictory statements that aren't well accounted for. In addition, overall, the citations are out of order.  I included some examples here;

Line 23; Remove; “Intracerebral hemorrhage (ICH) is commonly known . . . stroke type.” Instead tell the actual frequencies of ICH.  ICH is not a type of stroke.  A stroke can be hemorrhagic, but ICH is a general term for any bleeding from any cause – could be traumatic

Lime 33; Provide details of the burden. How many people annually?

Line 64; duplication of the word are

Line 67; Change “evidences have” to “evidence has”

Line 74; rewrite first sentence.

Line 88; “in” needs to go before the hippocampus

Line 99-103; Are you implying fluid cysts post hemorrhage is caused end-feet swelling?

Line 125-130; these contradictory statements need to more clear.

Line 132; Correct grammar

Line 132-139; This could be clearer – the flow of thoughts isn’t very clear

Line 154; “ICH-related brain damage involves mechanical destructive effect of hematoma and secondary brain injury” doesn’t make sense.

Line 248 – 254; The contradictory statements here aren’t cohesive.
